# Saline-Tolerant Pathogenic *Acanthamoeba* spp. Isolated from a Geothermal Power Plant

**DOI:** 10.3390/pathogens12111363

**Published:** 2023-11-17

**Authors:** Elizabeth Ramírez-Flores, Patricia Bonilla-Lemus, María M. Carrasco-Yépez, Miguel A. Ramírez-Flores, Karla A. Barrón-Graciano, Saúl Rojas-Hernández, María Reyes-Batlle, Jacob Lorenzo-Morales

**Affiliations:** 1Laboratorio de Microbiología Ambiental, CyMA, FES Iztacala, Universidad Nacional Autónoma de México, Avenida de los Barrios 1, Los Reyes Ixtacala, Tlalnepantla de Baz 54090, Mexico; blemus@unam.mx (P.B.-L.); macaye.tam@comunidad.unam.mx (M.M.C.-Y.); mikee2@comunidad.unam.mx (M.A.R.-F.);; 2Laboratorio de Inmunobiología Molecular y Celular, Estudios de Posgrado e Investigación, Escuela Superior de Medicina, Instituto Politécnico Nacional, Salvador Díaz Mirón S/N, Casco de Santo Tomás, Miguel Hidalgo 11340, Mexico; srojash@ipn.mx; 3Instituto Universitario de Enfermedades Tropicales y Salud Pública de Canarias, Universidad de la Laguna (ULL), Av. Astrofísico Francisco Sánchez S/N, 38206 Tenerife, Spain; mreyesba@ull.edu.es (M.R.-B.); jmlorenz@ull.edu.es (J.L.-M.); 4Departamento de Obstetricia y Ginecología, Pediatría, Medicina Preventiva y Salud Pública, Toxicología, Medicina Legal y Forense y Parasitología, Universidad de La Laguna, 38203 San Cristóbal de La Laguna, Spain; 5Centro de Investigación Biomédica en Red de Enfermedades Infecciosas (CIBERINFEC), Instituto de Salud Carlos III, 28220 Madrid, Spain

**Keywords:** pathogenic free-living amoebae, hot water, cooling system, conductivity, salinity

## Abstract

Few studies have been conducted in the cooling systems of power plants; they have focused on *Naegleria fowleri*, leaving a gap in the knowledge of other pathogenic free-living amoebae in this environment. The objective of this study was to determine the occurrence of saline-tolerant pathogenic *Acanthamoeba* in a geothermal power plant. The identification of isolated amoebae at genus level was carried out, observing their morphological characteristics; the determination of genotype and species of *Acanthamoeba* was performed via molecular biology (PCR). Water temperature ranged from 18 to 43 °C and conductivity from 4.0 × 10^4^ to 8.7 × 10^4^ μS/cm; this last value was greater than the seawater value. Only five amoeba genera were found. *Acanthamoeba* was in all the sampling sites, showing high saline tolerance. The high temperature, but mainly high conductivity, were the environmental conditions that determined the presence of pathogenic free-living amoebae in the hot water. All the strains of *Acanthamoeba culbertsoni* killed the mice, having a mortality of 40 to 100%. *Acanthamoeba* genotypes T10 and T5 were identified, T10 is rarely isolated from the environment, while T5 is more frequent. This is the first time that genotypes T5 and T10 have been reported in the environment in Mexico.

## 1. Introduction

Free-living amoebae (FLA) are widely distributed in the environment and have been found throughout the world. *Naegleria fowleri*, *Acanthamoeba* spp., *Balamuthia mandrillaris,* and *Sappinia pedata* can be pathogenic to humans and animals [1]. Pathogenic, and non-pathogenic FLA also have medical relevance because of their possible role as hosts or vectors for other pathogenic organisms [2,3]. *N*. *fowleri* causes a fatal infection of the central nervous system called Primary Amoebic Meningoencephalitis (PAM) in healthy people. Some species of *Acanthamoeba*, *Balamuthia mandrillaris*, and *Sappinia pedata* can cause a central nervous system infection called Granulomatous Amoebic Encephalitis (GAE). *Acanthamoeba* is an opportunistic amoeba that affects immunosuppressed individuals such as patients of AIDS; moreover, it can cause a corneal infection in contact lens wearers. This is aggravated by the lack of effective treatments [1]. 

Hot water is a suitable environment for thermophilic FLA [4,5,6,7,8,9]; they can tolerate temperatures over 30 °C. *Naegleria fowleri* and *Acanthamoeba* can grow well at high temperatures, even to 45 °C [5]. Thermophilic FLA can survive thanks to the resistance of their cyst or the potential to adapt to adverse conditions. Especially, *Acanthamoeba* can withstand extreme environmental conditions due to the presence of cellulose in the wall of their cyst [1]. Reports have mentioned that *Acanthamoeba* cysts can survive in extremely hostile environments; cysts stored at 4 °C can come out of the cyst even after 50 years [10]. Chlorine and pH can have an impact on FLA presence; these conditions could indirectly favor the presence of the more resistant amoebae such as *Acanthamoeba* [11]: these amoebae were isolated from hot sanitary water systems with a level of chlorine of 8 mg/L [12]. 

Geothermal power plants use the steam produced by geothermal energy inside the ground to move turbines and to produce electricity. As a result, hot water is produced by steam condensation; hence, a system to cool the hot water is required before the water is returned to the ground [13]. The water is saline, with high concentrations of chlorides, sodium, calcium, and potassium [14]. 

Salinity is a measure of the concentration of salts dissolved in water; salts dissolved in water decompose into positively and negatively charged ions. Conductivity measures the ability of water to conduct an electric current through dissolved ions. Therefore, salinity and conductivity are related; thus, when the concentration of dissolved ions increases, the values of both increase [15]. 

Conductivity has hardly been considered in studies of pathogenic FLA in aquatic environments [16,17,18], and few studies have been conducted on water with high salinity [19,20,21,22,23]. On the other hand, studies on the cooling systems of power plants have focused mainly on *Naegleria fowleri* [4,5,6,9], leaving a gap in the knowledge about the distribution of other FLA in this environment. For these reasons, the objective of this study was to determine the presence of saline-tolerant pathogenic *Acanthamoeba* in a geothermal power plant. 

## 2. Materials and Methods

### 2.1. Sampling

The study was carried out in the Geothermal Power Plant of Cerro Prieto; this is the second largest plant in the world, with an installed capacity of 820 MW and an area of 38.33 km^2^. The groundwater of the Cerro Prieto geothermal plant is extracted from 1500 to 3100 m deep geothermal wells that are in constant operation to generate electricity. The plant is in the Mexicali Valley, Baja California (Mexico) (Figure 1). The prevailing climate of the area is very warm and dry, with light rainfall in winter; in summer, the temperature can reach 50 °C [24]. The hot water produced during electric power generation is discharged into a cooling system that consists of secondary channels, a primary channel, and an evaporation lagoon. 

Samples of 500 mL each were collected from different sites at the plant: the Primary Discharge Channel (PDC), Secondary Discharge Channels (SDC), and Evaporation Lagoon (EvL) (Figure 2). Water samples were placed in sterile containers and kept at room temperature until analysis. Physicochemical parameters were measured in situ. Water temperature, conductivity (K_25_), and pH were measured using a conductimeter model PC18 (Conductronic Instruments, Puebla, Mexico); dissolved oxygen was measured with an oxymeter YSI Model 51 B. These analyses were carried out using standard methods [15].

### 2.2. Processing Samples

Water samples were homogenized and filtered through a 2.0 µm-diameter Millipore filter (Millipore, Bedford, MA, USA); membranes were placed on Petri dishes with a non-nutrient agar medium with *Enterobacter aerogenes* ATCC-13048 (NNA). This procedure was carried out in duplicate to incubate at 30 °C and 37 °C. NNA plates were examined through an inverted microscope (Zeiss model 473028) to detect the growth of free-living amoebae.

Amoebae were sub-cultivated in a fresh NNA medium several times to clean and separate them from other microorganisms. Blocks (1 cm^2^) were taken from the NNA medium containing trophic amoebae and transferred to axenic culture, modified Chang medium, and 2% (*w*/*v*) Bacto™Casitone medium (Difco, Le Pont de Claix, France), supplemented with 10% (*v*/*v*) fetal bovine serum (Gibco, Grand Island, NY, USA) and 100 U penicillin and 100 µg of streptomycin per milliliter. Cultures were incubated again at 30 °C or 37 °C to obtain pure isolates of the amoebae [25].

### 2.3. Morphological Identification

The traditional identification based on the morphological characteristics of FLA was used to identify the amoebae at genus level, and molecular biology techniques were used to identify at species level the pathogenic FLA. In vivo preparations of the amoebae from axenic and monoxenic cultures were examined through a phase-contrast optical microscope at magnifications of 40× and 100× (Zeiss K7, Germany). The morphological identification was made, considering the trophozoite and cyst characteristics of amoebae using taxonomical keys for FLA [26].

### 2.4. Pathogenicity Test

Pathogenicity tests using mice were in accordance with the Mexican federal regulations for animal experimentation and care (NOM-062-ZOO-1999, Ministry of Agriculture, Mexico City, Mexico), and approved by the ethical standards of the Facultad de Estudios Superiores Iztacala, UNAM (Number of Approval CE/FESI/022020/1317). In all experiments, 6- to 8-week-old male BALB/c mice were used. Five mice were maintained per cage and fed ad libitum. This test was carried out on mice using axenic amoeba cultures. Trophozoites were concentrated at 3000 rpm for 10 min., and were adjusted to a concentration of 1 × 10^5^ per ml. A volume of 0.02 mL was taken and was inoculated intranasally. Five mice were inoculated with an amoeba-free culture medium for controls. Mice that were about to die, and that survived after the 21 days, were euthanized; these and mice that died during the test period were opened to remove their brain, liver, lungs, and kidneys and placed on plates with non-nutrient agar medium that was incubated at isolation temperature (30 or 37 °C). Cultures were observed every day for a week to monitor the development of amoebae. The mortality rate was determined, taking in account the dyed mice [27].

### 2.5. DNA Isolation and PCR

Genomic DNA was extracted from those morphologically identified isolates such as *Acanthamoeba*. Amoebic cultures were placed directly into a Maxwell 16 tissue DNA purification kit sample cartridge (Promega, Madrid, Spain). The amoebic genomic DNA was purified using the Maxwell 16 instrument in according to the manufacturer recommendations. DNA yield and purity were determined with a NanoDrop spectrophotometer (DeNovix^®^, USA DS-11, Wilmington, DE, USA).

Polymerase Chain Reaction (PCR) was conducted as previously described [28] where Genus-specific PCRs were run for *Acanthamoeba* using the following primers: JDP1 (5′-GCCCAGATCGTTTACCGTGAA-3′) and JDP2 (5′-TCTCACAAGCTCTAGGGAGTCA-3′). Depending on the genotype, the primers amplify 423 to 551 bp of the 18S ribosomal DNA (rDNA) between reference bp 936 and 1402.

A mix PCR was prepared with 50 µL volume, containing 1.25 U Taq DNA polymerase (Ecogen), 40–100 ng DNA, 1.5 mM MgCl_2_, 200 µM deoxynucleoside triphosphate (dNTP), and 0.2 µM each primer.

Once the mix had been prepared, samples were placed in an Artik Thermal Cycler (Thermo Scientific, Waltham, MA, USA) and run under the following conditions: an initial 5 min denaturation at 95 °C, followed by 35 cycles of amplification at 95 °C, with 30 s denaturing at 95 °C, 30 s annealing at 60 °C, 30 s extension at 72 °C, and a final extension of 7 min at 72 °C. 

Amplicons (5 μL aliquot of the PCR reaction) were analyzed by electrophoresis on 2% agarose gel (SIGMA, St. Louis, MO, USA) in TBE buffer, stained with Safe-Green. The gel was then displayed on a UV transilluminator-Gel Doc XR (Bio-Rad) at a wavelength of 260 nm. 

### 2.6. Sequencing and Phylogenetic Analysis

The PCR products of isolates were then used for sequencing, using an automated fluorescent sequencing system (Macrogen Spain service, Av. Sur del Aeropuerto, Madrid, España). The obtained sequences were aligned using the Mega 3.0 software program. *Acanthamoeba* genotype identification was based on sequence analysis of the DF3 region as previously described and the comparison to the available DNA sequences in GenBank.

## 3. Results

### 3.1. Physicochemical Parameters

The temperature distribution in the cooling system presented a cooling in accordance with the path of water circulation (secondary channels, primary channel, and evaporation lagoon). This behavior had been reported before in cooling systems [29]. The temperature was lowered from 43 to 18 °C (Table 1), fulfilling the objective of the cooling process. The water in the discharge channels (SDC and PDC) had a conductivity of 4.0 to 4.3 × 10^4^ µS/cm, and in the Evaporation Lagoon (EvL) it increased to 8.7 × 10^4^ μS/cm. The pH of the water was nearly neutral, and the values of dissolved oxygen ranged from 1.8 to 5.0 mg/L (Table 1).

### 3.2. Presence of Free-Living Amoebae in the Sampling Sites 

Five genera of amoebae were found in the hot water. *Acanthamoeba* was present at all the sites, followed by *Vannella* and *Vermamoeba*, while *Saccamoeba* and *Thecamoeba* were only in one site (Table 2).

### 3.3. Morphological Description of Acanthamoeba

*Acanthamoeba* isolates were identified at the genus level via morphological criteria; all the isolated strains belonged to *Acanthamoeba* group III [26]. The trophozoite of *Acanthamoeba* presents fine pseudopods called acanthopods, which gives the amoeba a “thorny” appearance. It has a spherical nucleus with a central nucleolus that occupies much of the nucleus. A large contractile vacuole and several digestive vacuoles are usually seen in the cytoplasm. The cyst of *Acanthamoeba* species has two layers on its wall, the ectocyst and the endocyst. In morphological group III, the ectocyst is a thin layer and the endocyst is rounded without forming arms. The two layers are close together around most of the circumference (Figure 3).

### 3.4. Morphological and Molecular Biology Identification of Acanthamoeba Strains 

Based on molecular techniques, four strains (SDC1, SDC2a, PDC, and EvL1) were identified as *Acanthamoeba culbertsoni* belonging to genotype T10, and two strains (SDC2a and EvL2) were identified as *Acanthamoeba lenticulata* belonging to genotype T5 (Table 3).

### 3.5. Pathogenicity Test

All the strains of *A*. *culbertsoni* were pathogenic in mice, presenting a mortality of 40 to 100%. SDC2a had the highest mortality (100%), killing mice in a few days; SDC1 and PDC had 60% and 80% mortality, respectively, killing the mice in a few further days; and EVL1 had the lowest mortality (40%), killing the mice in more days. In contrast, only one of the strains of *A*. *lenticulata* had 20% mortality, and the other did not kill any mice. The optimal growth temperature of pathogenic amoebae of *A*. *culbertsoni* was 37 °C, while the non-pathogenic amoebae of *A*. *lenticulata* grow better at 30 °C (Table 4).

Macroscopic damage was observed in the brain and lung of dead mice, caused by infection with *Acanthamoeba culbertsoni*. It was proven that the infection was caused by amoebae by inoculating a sample of brain and lung from dead mice on NNA medium, where the amoebae were recovered, and by observing rounded trophozoites in the brain tissue in a brain imprinting from a dead mouse (Figure 4). 

## 4. Discussion

The presence of some FLA in the hot water of the Geothermal Power Plant indicates the endurance of amoebae at high temperatures. It was observed that *Acanthamoeba*, *Vermamoeba,* and *Vannella* could grow at high temperatures up to 43 and 45 °C [3,11,30,31]. Thermotolerant *Acanthamoeba* and *Vermamoeba* (previously *Hartmannella*) had already been isolated from the cooling lagoon of an electric power plant [5]. The capacity of these amoebae to grow at high temperatures explains their frequency in the Geothermal Power Plant of Cerro Prieto. On the other hand, it was a surprise to find amoebae of the genera *Saccamoeba* and *Thecamoeba*, because they had not been reported in hot water.

The pH was alkaline in the water of the channels and was slightly acid in the evaporation lagoon. This is due to the underground origin of the water [14]. Dissolved oxygen was in a range of 1.8 to 5.0 mg L^−1^; these values were appropriate for the presence of the pathogenic FLA [1].

Geothermal water contains many salts, often in varying concentrations [14]. The evaporation lagoon functions as a temporary reservoir for water extracted from the geothermal reservoir during the generation of electrical energy. The water that circulates through the evaporation lagoon undergoes a continuous evaporation until reaching a maximum average value of 58%. In semi-arid climatic conditions, evaporation plays an important role in the concentration of the salts in water [29]. Thus, the ions in the lagoon water, and therefore the conductivity, showed a great increase in the channels to the evaporation lagoon. The conductivity of the water in the discharge channels was near to that of seawater (5.0 × 10^4^ μS/cm) [32] and was higher in the evaporation lagoon. The values of conductivity were much higher than reported in the irrigation channels (average of 1.5 × 10^3^ μS/cm) [17] and swimming pools (average of 2.4 × 10^3^ μS/cm) [16,18].

The high conductivity of the water restricted the presence of the pathogenic free-living amoebae, only five genera of amoebae were found. *Acanthamoeba* was found at all sites, even in the evaporation lagoon with the highest conductivity. Thus, it showed great tolerance to high concentrations of salts, and reaffirmed its high resistance to extreme conditions [10,11]. 

It is important to highlight the absence of *Naegleria fowleri* in the hot water of the geothermal power plant, being a thermophilic amoeba. This amoeba has been reported in different sites with hot water (a thermally polluted channel, geothermal spring, cooling pond of an electric power plant, cooling reservoir of a nuclear power plant, geothermal recreational baths, and cooling lake of a nuclear reactor) [4,5,6,7,8,9]. However, the high conductivity of the water did not allow the growth of the amoeba. It has been reported that trophozoites of *N*. *fowleri* are destroyed in saline concentrations >2% [33], equivalent to 3.1 × 10^4^ µS/cm of conductivity, which is lower than the water conductivity of the geothermal power plant.

*Acanthamoeba* genotype T10 was the most identified in our isolates, followed by genotype T5. These two genotypes were reported in recreational and domestic water sources in Jamaica and West Indies [34]. Genotype T10 is rarely isolated from the environment and is usually associated with Granulomatous Amoebic Encephalitis, while genotype T5 is more frequently isolated from the environment [35]. This is the first time that *Acanthamoeba* genotypes T5 and T10 have been reported in the environment in Mexico. Four strains were classified as genotype T10 *Acanthamoeba culbertsoni*, and two strains belonged to genotype T5 *Acanthamoeba lenticulata*. This last amoeba had already been isolated from the cooling lagoon of an electric power plant [5]. 

The strains of *A*. *culbertsoni* showed a gradient of pathogenicity, from the least pathogenic, with 40% mortality, to the most pathogenic, with 100% mortality; this behavior had also been observed in *A*. *polyphaga* strains in swimming pools [16]. This is significant, because the treated water from the Geothermal Power Plant is returned to the ground and can pollute the aquifer of Mexicali Valley, due to that one part of the aquifer being free, which allows the infiltration to groundwater [36]. The pathogenic amoebae of *A*. *culbertsoni* were thermotolerant, and grew well at 37 °C and 42 °C. This agrees with the assertion that pathogenic amoebae are thermophilic [1].

*Acanthamoeba* is widespread in nature. In Mexico, free-living amoebae (FLA) have been isolated from different environments such as air, thermal waters, physiotherapy tubs, groundwater, tap water, rivers, wastewater, springs, swimming pools, and irrigation channels [16,17,25,37,38,39,40,41,42,43].

The *Vermamoeba* genus, with only one species, *Vermamoeba vermiformis* (previously described as *Hartmannella vermiformis*), has been a subject of increased interest in recent times. This amoeba is one of the most prevalent, ubiquitous, and thermotolerant [3,30,31]. It has been isolated from different aquatic environments [16,39,42,43]. In groundwater and thermal springs, it was more frequently found than *Acanthamoeba* and *Naegleria* [39,43] and in textile wastewater, it was almost in the same frequency as *Acanthamoeba* [42]. There is no clear evidence that it can cause brain infections. The only relationship of *Vermamoeba vermiformis* with a brain infection was a case of GAE, but the role of the amoeba as a causative agent of the disease could not be proven [44]. However, *V*. *vermiformis* has demonstrated an ability to produce a cytopathic effect on keratocytes in vitro [45]. The first case of amoebic keratitis caused by *V*. *vermiformis* was reported in a soft contact lens wearer in Iran, and there have been reports of cases of keratitis due to *V*. *vermiformis* in conjunction with other amoebae [46]. Furthermore, it is known that *V*. *vermiformis* can harbor viruses, bacteria such as *Legionella pneumophila*, and other pathogen microorganisms [3,31]. 

*Vannella* is commonly found in aquatic environments such as freshwater, wastewater, tap water, groundwater, springs, and swimming pools [16,39,40,41,42,43,47]. *Vannella* has not been reported as a pathogen but was isolated from a keratitis patient. It is a natural host and vehicle of transmission for various intracellular organisms. This amoeba can act as a Trojan horse for microsporidian parasites and other pathogens [48].

*Saccamoeba* is usually found in soil and in different aquatic environments such as groundwater, streams, and wastewater [39,41,42]. However, some species (*S*. *marina*, *S*. *osseosaccus* and *S*. *verrucosa*) were found in marine water [20,49]. *Saccamoeba* have not been reported to cause disease, but *S*. lacustris was found harboring mutualistic rod-shaped gram-negative bacteria [50].

*Thecamoeba* is present in fresh water, salt water, and wastewater, and is among the largest protozoa in soil. *Thecamoeba* feed on a variety of protozoa and algae, as well as bacteria, and occupy an important ecological position amongst the microbiota of both aquatic and terrestrial habitats [39,41,42]. It has not been reported as pathogenic. 

The presence of some species of *Saccamoeba* and *Thecamoeba* in marine water might be the reason that were found in the Geothermal Power Plant. 

## 5. Conclusions

The high temperature, but mainly high conductivity, were the environmental conditions that determined and restricted the presence of the pathogenic free-living amoebae in the Geothermal Power Plant, finding that few amoeba genera were tolerant to salinity. *Acanthamoeba* was the only amoeba found at all sites, even in the evaporation lagoon with the highest conductivity, confirming its resistance to extreme environmental conditions. *Vannella* and *Vermamoeba* were at several sites in the cooling system, also showing tolerance to high temperatures and salinity. The strains of *Acanthamoeba culbertsoni* were pathogenic; it is an opportunistic amoeba that may affect immunosuppressed individuals; this is significant because the treated water is returned to the ground and can pollute the aquifer. This is the first time that *Acanthamoeba* genotypes T5 and T10 have been reported in the environment in Mexico. 

## Figures and Tables

**Figure 1 pathogens-12-01363-f001:**
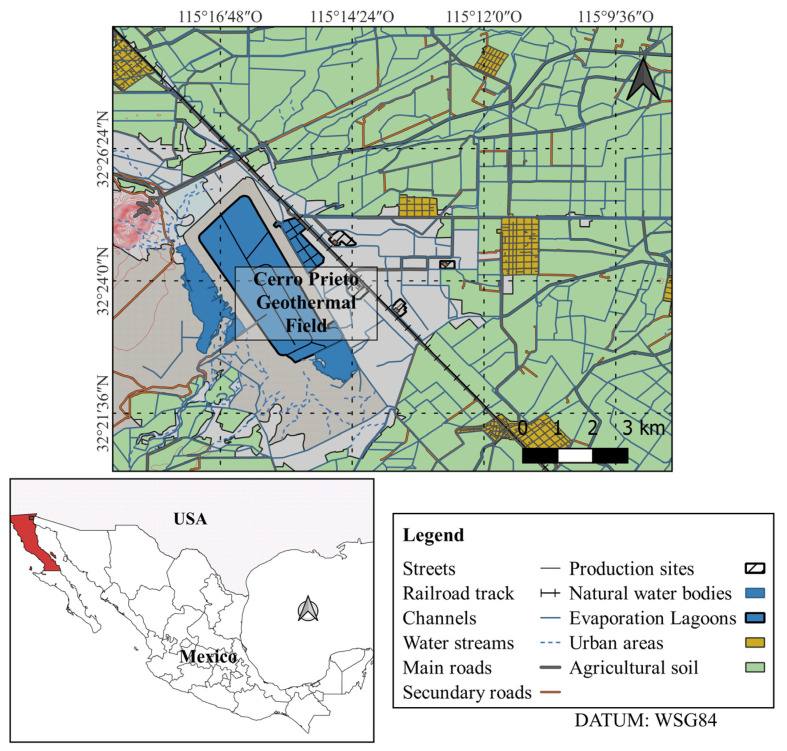
Localization of Cerro Prieto Geothermal Power Plant, Baja California, Mexico.

**Figure 2 pathogens-12-01363-f002:**
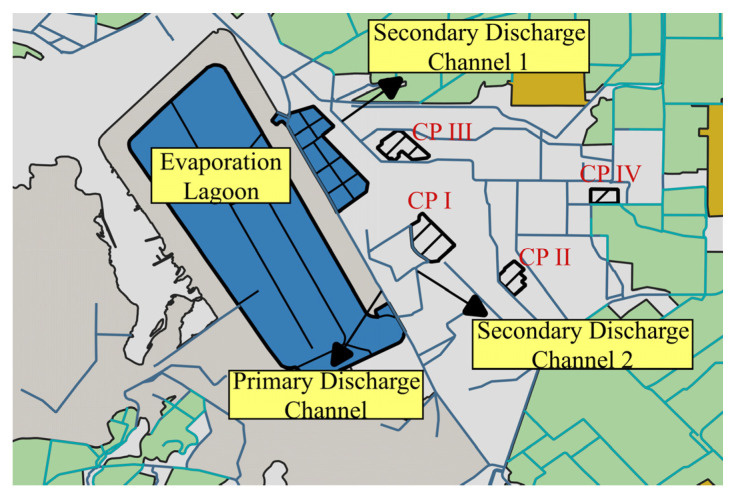
Sampling sites at Cerro Prieto Geothermal Power Plant. Yellow boxes, sampling sites; blue lines, water channels; CPI, CPII, CPIII, and CPIV production units.

**Figure 3 pathogens-12-01363-f003:**
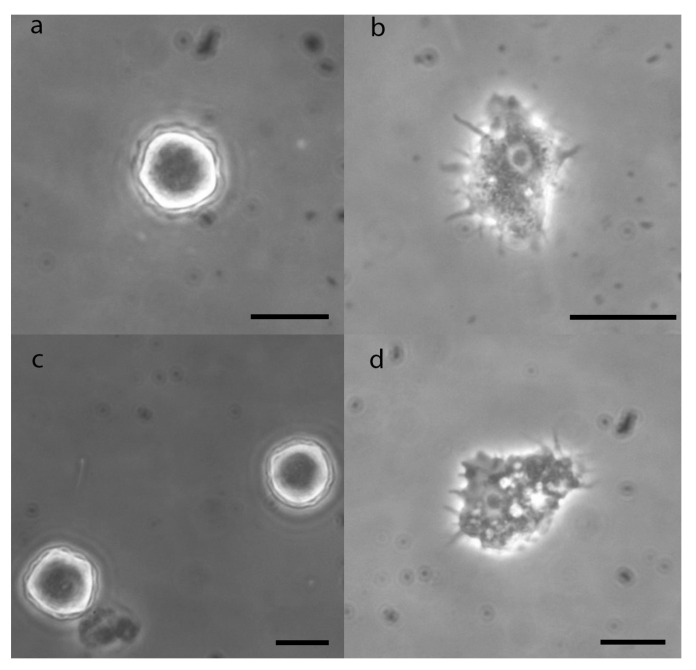
Micrographs of *Acanthamoeba* isolates. *A*. *lenticulata*: (**a**) cyst, (**b**) trophozoite. *A*. *culbertsoni*: (**c**) cyst, (**d**) trophozoite. Bar: 10 µm. Phase-contrast microscopy.

**Figure 4 pathogens-12-01363-f004:**
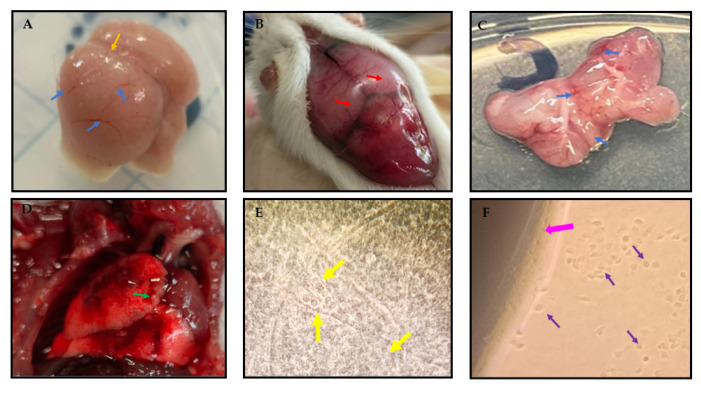
Macroscopic aspects of damage caused by *Acanthamoeba culbertsoni* infection in mouse brain and lung. (**A**) Observations were made of the inflammatory process of the brain, the loss of integrity of the central sulcus (orange arrow), and areas with the beginnings of hemorrhagic processes (blue arrows) in a dying mouse brain euthanized 4 days after infection. (**B**) Mouse brain dead 5 days after infection showed hyperemic meninges (red arrows) that stand out in dorsal view oriented from the central sulcus of the brain and towards the posterior region of the cerebellum. (**C**) Mouse brain recovered 2 to 4 h after mouse death. Observations were made that the integrity of the brain has been totally lost, the bleeding areas and edema are very evident (blue arrows). (**D**) Mouse lung dead 12 days after infection; there was abundant hyperemic zone in the anterior lobe; the hemorrhage extended to the central area and in the periphery, and presence of small continuous granules was observed (green arrow). (**E**) Rounded trophozoites of amoebas (yellow arrows) immersed in brain tissue observed in brain imprinting from a dead mouse. Optical microscope at 40×. (**F**) Amoeba trophozoites (purple arrows) coming out of the brain of a dead mouse (pink arrow) inoculated on NNA, after 24 h of culture at 37 °C. 10× inverted microscope.

**Table 1 pathogens-12-01363-t001:** Physicochemical parameters of sampling sites.

Sampling Sites	Site ID	pH	Water Temperature (° C)	Dissolved Oxygen (mg/L)	Conductivity (µS/cm)	* Salinity (%)
Secondary Discharge Channel 1	SDC1	7.9	42.5	2.0	4.3 × 10^4^	2.75
Secondary Discharge Channel 2	SDC2	8.1	43.0	1.8	4.2 × 10^4^	2.68
Primary Discharge Channel	PDC	7.3	34.0	2.2	4.0 × 10^4^	2.56
Evaporation Lagoon 1	EvL1	6.6	22.0	5.0	4.1 × 10^4^	2.62
Evaporation Lagoon 2	EvL2	6.7	18.0	5.0	8.7 × 10^4^	5.56

* Conductivity was converted to percentage of salt concentration using Lenntech online converter and the equivalence of Lam.

**Table 2 pathogens-12-01363-t002:** Free-living amoebae isolated from Geothermal Power Plant.

Site	Genus
SDC1	*Acanthamoeba* (Volkonsky 1931) *Vannella* (Bovee 1965) *Vermamoeba* (Cavalier-Smith and Smirnov 2011)
SDC2	*Acanthamoeba* (Volkonsky 1931) *Saccamoeba* (Frenzel 1982 emend. Bovee 1972) *Thecamoeba* (Fromentel 1874) *Vannella* (Bovee 1965)
PDC	*Acanthamoeba* (Volkonsky 1931) *Vannella* (Bovee 1965) *Vermamoeba* (Cavalier-Smith and Smirnov 2011)
EvL1	*Acanthamoeba* (Volkonsky 1931)
EvL2	*Acanthamoeba* (Volkonsky 1931)

**Table 3 pathogens-12-01363-t003:** Morphological and molecular biology identification of *Acanthamoeba* strains isolated from Geothermal Power Plant.

Strain	Morphology	Molecular Biology (JDP1/2)		
		Species	Genotype	GenBank Accession Number
SDC1	*Acanthamoeba* Group III	*Acanthamoeba culbertsoni*	T10	OR767829
SDC2a	*Acanthamoeba* Group III	*Acanthamoeba culbertsoni*	T10	OR767828
SDC2b	*Acanthamoeba* Group III	*Acanthamoeba lenticulata*	T5	OR767826
PDC	*Acanthamoeba* Group III	*Acanthamoeba culbertsoni*	T10	OR767830
EvL1	*Acanthamoeba* Group III	*Acanthamoeba culbertsoni*	T10	OR767831
EvL2	*Acanthamoeba* Group III	*Acanthamoeba lenticulata*	T5	OR767827

**Table 4 pathogens-12-01363-t004:** Pathogenicity and temperature tolerance of *Acanthamoeba* strains isolated from Geothermal Power Plant.

Strain	Mortality (%)	Death Days	Organs from FLA Were Recovered	Temperature Tolerance (°C)
SDC1	60	15–17	Brain and Lung	30, 37 *, 42
SDC2a	100	2–11	Brain	30, 37 *, 42
SDC2b	20	16	Brain and Lung	30 *, 37, 42
PDC	80	12–13	Brain and Lung	30, 37 *, 42
EvL1	40	18–20	Lung	30, 37 *, 42
EvL2	0	-	-	30 *, 37, 42

* Optimal growth temperature.

## Data Availability

Research data are included in the article.

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
