# Peer review of "Saline-Tolerant Pathogenic Acanthamoeba spp. Isolated from a Geothermal Power Plant"

_pathogens, 2023, doi:10.3390/pathogens12111363_

Round 1

Reviewer 1 Report

Comments and Suggestions for Authors

Title: Saline-tolerant pathogenic free-living amoebae isolated from a geothermal power plant

This manuscript is concise and shows clear description for results.

 1. As the results, since the only description is about Acanthamoeba among the free-living amoeba that were isolated and cultured, it would be good to change the title to more subdivided, for example, “Saline-tolerant pathogenic Acanthamoeba spp. isolated from a geothermal power plant”.

2. Also, the genus-specific PCR primers (JDP1, JDP2) were used, but I think it would have been better if there had been an experiment using species-specific primers.

3. To observe the pathogenicity of A. culbertsoni, the results of experiments using a mouse animal model were very good. In addition, it would be nice to show the cultured trophozoites or pictures of brain and lung tissues from mice infected with Acanthamoeba.

Reviewer 2 Report

Comments and Suggestions for Authors

In the manuscript „Saline-tolerant pathogenic free-living amoebae isolated from a geothermal power plant” the authors investigated water samples from several different sampling sites for the presence of free-living amoebae. They isolated in my opinion “the usual suspects”. What makes it new and interesting is the fact, that they found several Acanthamoeba strains, representing genotype T10, usually associated with brain infections and the high salinity and high temperatures present at the sampling sites. They showed that these strains all exhibited pathogenic potential in mice and indicated a potential health risk, since waters from the power plant might contaminate ground water.

General remarks

The study is interesting and the methodology is sound, however the manuscript needs significant improvement.

In general, it felt kind of “sloppy”. There are many unnecessary mistakes, starting in the abstract. The Englisch language must be improved, in particular in the discussion section, maybe with the help of a native speaker.

The introduction but more so the discussion section could be shortened and should be more concise and some citations are probably not essential. A lot of facts are provided several times throughout the manuscript, which should be avoided.

It is not suitable for publication in its current form, but might be an interesting contribution for the scientific community after careful revision. Also I hope to get some of my questions answered!

Specific remarks:

P4line144: Were all the surviving mice euthanized after 21 days? I am not familiar with mice experiments. What does “euthanized when they died” mean? Were they killed, when it was apparent that they would die?

P5 line 167: Is this correct? 50°C annealing is too low for this PCR. Usually, this program would run with 60°C to be specific.

P5 line 177: It this citation essential?

Table 1 and 2: Maybe these tables should be combined to have the parameters and respective amoebae together, or at least table 2 should be altered to make it clearer, with lines separating the sampling sites.

P6, line 1944f: This could be shortened. If Acanthamoeba are shown in such detail here, the other amoebae should be too, since they were only identified by morphology, while acanthamoebae were also identified by PCR.

P6, line 208f: Please upload your sequences to genbank and provide accession numbers! Since T10 is not that common, it would be an important contribution.

Table 4: It is interesting to see that also EvL1 and EvL2 had an optimum growth at 30 and 37°C, although they were found in samples with very low temperatures. In my experience many strains do not grow at 37°C and definitely not at 42°C (even strains from patients). Did they all grow at 42°C?

P7 line 232: I don’t know about Saccamoeba but Thecamoeba have an affinity for “salty” conditions. I find the fact, that these two amoebae have been reported from hot water for the first time (if that is correct) quite interesting, although they were not shown to be associated with diseases. Maybe some more info would be interesting here.

P7 line 242: What does 58% maximum average value mean? 58% of salts?

P7 line 243f: I understand that higher conductivity equals higher salt concentration, however, can it be translated into a percentage?

P8 line 261: Here a saline concentration of 2% is given, how can the reader compare 2% to a conductivity of 4.3 x104?

P8 line262: It should be mentioned that you are writing about Acanthamoeba here.

P8 line 279f: This has already been stated in the introduction.

P8 line 283: again, how can 8mg/L be compared to what you measured in your study?

P8 line 289f: The last two paragraphs could be shortened. 

Comments on the Quality of English Language

already state before, should be improved!
